# A New Xanthone Glycoside from *Mangifera indica* L.: Physicochemical Properties and In Vitro Anti-Skin Aging Activities

**DOI:** 10.3390/molecules27092609

**Published:** 2022-04-19

**Authors:** Heba A. S. El-Nashar, Eman M. El-labbad, Mahmood A. Al-Azzawi, Naglaa S. Ashmawy

**Affiliations:** 1Department of Pharmacognosy, Faculty of Pharmacy, Ain Shams University, Cairo 11566, Egypt; naglaa.saad@pharma.asu.edu.eg; 2Centre of Drug Discovery Research and Development, Ain Shams University, Cairo 11566, Egypt; 3Department of Pharmaceutical Sciences, College of Pharmacy, Gulf Medical University, Ajman P.O. Box 4184, United Arab Emirates; 4Pharmaceutical Chemistry Department, Faculty of Pharmacy, Ain Shams University, Cairo 11566, Egypt; 5Department of Medical Laboratory Technologies, Al-Amal University College for Specialized Medical Sciences, Karbala P.O. Box 56001, Iraq; mmahmood41@yahoo.com; 6Rochester Institute of Technology-Dubai, Dubai P.O. Box 341055, United Arab Emirates

**Keywords:** xanthone, *Mangifera indica*, aging, collagenase, elastase, tyrosinase, hyaluronidase, in silico ADMET

## Abstract

A new xanthone glycoside, 1,3,5,6-tetrahydroxyxanthone-C-4-β-d-glucopyranoside was isolated from the methanol extract of *Mangifera indica* leaves (Anacardiaceae) growing in Egypt. The structure was clarified by 1D and 2D-NMR spectroscopic data. The physicochemical properties of the compound such as lipophilicity, solubility, and formulation considerations were predicted via in silico ADMET technique using the SwissADME server. This technique provided Lipinski’s rule of five, such as GIT absorption, distribution, metabolism, and skin permeation. The in vitro inhibitory activities against aging-mediated enzymes such as collagenase, elastase, hyaluronidase, and tyrosinase were assessed. The compound exhibited remarkable anti-collagenase, anti-elastase, anti-hyaluronidase, and anti-tyrosinase effects with IC_50_ values of 1.06, 419.10, 1.65, and 0.48 µg/mL, respectively, compared to the positive control. The compound showed promising predicted aqueous solubility and reasonable skin penetration suggesting the suitability of the compound for topical formulation as an anti-aging agent for cosmetic preparations.

## 1. Introduction

The aging process is a complex biochemical process associated with oxidative stress driven by endogenous oxygen and nitrogen free radicals generated during the life expectancy, which may contribute to the progression of age-associated manifestations [1]. Skin-aging pathologies such as wrinkles, hyperpigmentation, aging spots, wrinkles, melasma, freckles, lentigo, ephelides, nevus, browning, and melanoma are proceeded by activation of reactive oxygen species (ROS) [2]. Free radicals or ROS could potentially induce a change in structural skin cell composition and damage the cell membranes by stimulating the oxidation of lipids and proteins, leading to DNA damage and cell death [3,4]. Moreover, ROS play a significant role in the skin aging process by damaging major skin proteins such as collagen and elastin by activation of collagenase and elastase enzymes. Moreover, free radicals cause hyaluronic acid degradation through the activation of hyaluronidases, which leads to improper hydration of the skin [5,6]. Anticipation of these dynamic processes is considered a key issue for the dermo-cosmetics area and substantial research labor is being expensed to discover new protective formulations [7]. Antioxidants are the natural defense enzymes present in the skin and counteract the excessive free radicals produced in the body. However, the oxidative balance may be disrupted by several factors including diet and air pollution which lead to oxidative stress. In this oxidative stress, the load of free radicals in the body is significantly higher than natural antioxidants. Therefore, it is highly important to use external antioxidants from sources such as diet, cosmetics, or pharmaceuticals to neutralize free radicals and counter the skin aging process [5,8]. Modern studies have revealed that several natural secondary metabolites, mainly flavonoids and polyphenolics, contribute significantly to the total antioxidant activity of many plants [9]. This has led to the interest in the investigation of the antioxidant activity of many plants [10]. Fruits have been reported for their richness in these phenolics and flavonoids as well as vitamins [11]. 

Elastin constitutes the elastic protein of epidermal connective tissues with collagen and represents a substantial role in the prevention of skin wrinkles and firmness [12]. During cell aging, matrix metalloproteinases such as collagenase, elastase, and hyaluronidase are upregulated [13]. The suppression of these enzymes is one of the most effective therapeutic strategies to manage the deterioration of skin conditions during aging [14]. Plant-derived phenolic compounds have been described as potent whitening mediators due to their tyrosinase inhibitory property [15] and some of them could play a role in matrix remodeling via an elastase inhibitory effect [16].

*Mangifera indica* L. is one of the most important edible plants that belongs to the family Anacardiaceae distributed in many countries, particularly tropical regions [17]. Mango fruits have exhibited an imperative role in the agricultural, food, pharmaceutical, and nutraceutical fields for over 4000 years [18]. Moreover, mango fruit stands in the 2nd position among the industrial crops, after banana, in terms of production and coverage [19]. Different parts of *Mangifera indica* represent a wealthy source of different phytoconstituents including flavonoids, xanthonoids, phenolic acids, and triterpenoids with potential value as functional molecules [18]. In addition, fruits supply the human body with vital components such as carbohydrates, proteins, fats, minerals, vitamins, essential amino acids, carotenoids, dietary fibers, and phenolics [20]. Traditionally, different cultures reported that leaf infusion has been utilized for several ailments such as diarrhea, bloody dysentery, anemia, asthma, bronchitis, hypertension, insomnia, rheumatism, indigestion, hepatitis, tetanus, miscarriage, and hemorrhage [21]. Further, burning leaves’ fumes were inhaled for hiccups and throat affections [18]. The bark was used as a diuretic and in antirheumatic dressings. The seeds are well known for the treatment of colds, asthma, diarrhea, and bleeding piles. Antioxidant activity, high total phenolic (TPC), total flavonoid contents (TFC), 2,2-diphenyl-1-picrylhydrazyl (DPPH) scavenging activity, and linoleic inhibition capacity of *Mangifera indica* plant parts have been reported [22].

This study was designed to investigate the in vitro inhibitory properties of a new xanthone compound obtained from the 70% methanolic extract of *Mangifera indica* L. leaves against collagenase, elastase, tyrosinase, and hyaluronidase. Further, pharmacokinetic properties such as absorption, distribution, metabolism, and toxicity (in silico ADMET prediction) were conducted using SwissADME and Lipinski’s rule of five to assess the drug solubility, permeability, and formulation necessities.

## 2. Results and Discussion

### 2.1. Structure Elucidation of the Isolated Compound

The compound was obtained as yellow crystals (18.29 mg). Chromatographic analysis revealed a pale-orange yellow spot on TLC under long UV light (365 nm) in DCM:MeOH (7:3) and BAW solvent systems. When the spot was treated with 1% methanolic ferric chloride solution, it turned dark green and gave a yellow with spraying ammonia solution. The ^1^H- NMR, APT, and HMBC spectral data of the compound are compiled in Table 1. ^1^H-NMR spectral analysis (DMSO-*d_6_*, room temperature) (Appendix A) revealed the presence of two downfield aromatic doublet signals at δ_H_ 7.57 and 6.79 with a coupling constant of 8.24 Hz indicating the presence of two ortho-coupled protons. Moreover, the aromatic region of the ^1^H-NMR spectrum showed the presence of a singlet signal at δ_H_ 5.96, indicating the presence of a single proton in the aromatic ring. The anomeric proton of the C-glucosyl unit was detected at δ_H_ 4.60 ppm with a coupling constant value of 7 Hz assigned for β-anomer. This pattern of aromatic protons in the ^1^H-NMR spectrum of the isolated compound resembles the xanthone structure. Furthermore, the presence of a β-anomeric proton at δ_H_ 4.60 with characteristic chemical shifts of glucose at 3.21 to 4.60 of ^1^H-NMR (sugar region) indicated that there is a C-glycosidic linkage between the sugar and xanthone. As shown in Appendix A, the APT spectrum of the compound revealed the presence of nineteen carbon signals, representing the xanthone and the C-glucoside structure. The APT experiment identified three CH groups (C-8, 131.95; C-7, 115.10; C-2, 95.35 ppm) in the xanthone moiety and the anomeric C-1’ of the C-glucose at δ_H_ 75.13 ppm. We conducted two-dimensional NMR experiments (HMBC; Appendix A) to confirm the glycosidic linkage of the C-glucose at C-4 of the xanthone. Abundant 2,3JCH long-range correlation signals corroborated the xanthone backbone with observed cross-peaks from H-1’ (δ_H_ 4.60) to C-4 (δ_C_ 104.13 ppm) and C-3 (δ_C_ 158.95). For the xanthone moiety, a 2,3JCH long-range correlation was also detected from H-2 (δ_H_ 5.96) to C-4 (δ_C_ 104.13 ppm), C-3 (δ_C_ 158.95), and C-4b (δ_C_ 107.45). Moreover, there was a ^2,3^J^CH^ long-range correlation detected from H-8 (δ_H_ 7.57) to carbonyl group C-9 (δ_C_ 195.14), C-8a (δ_C_ 131.22), and C-6 (δ_C_ 161.87). For the same ring, ^2,3^J_CH_ signals from H-7 (δ_H_ 6.79) to C-8a (δ_C_ 131.22) were observed. These correlations are illustrated in the structure of the compound in Figure 1. HSQC of the compound showed _1_J_CH_ correlations between each proton and its located carbon as shown in Appendix A. The ^1^H,^1^H COSY experiment (Appendix A) revealed a strong cross-peak correlation between H-7 (δ_H_ 6.79) and H-8 (δ_H_ 7.57). Further, the compound revealed a pseudomolecular ion peak [M-H]^−^ at *m/z* 421, indicating a molecular formula of C_19_H_18_O_11_ in the negative mode of the ESI-MS analysis (Appendix A). The data from 1H-NMR, APT, ^1^H,^1^H COSY, HMBC, and HSQC led us to postulate that the compound could be 1,3,5,6-tetrahydroxyxanthone-C-4-β-D-glucopyranoside. This was confirmed by comparison with NMR data reported of isomangiferin in the literature [23,24]. 

### 2.2. In Silico Pharmacokinetics Prediction of the Isolated Compound

The SwissADME online server was used to assess the drug ability of the isolated compound by estimating Lipinski’s rule of five (RO5) for drug-likeness [25]. Lipinski reported that 90% of orally active drugs that reached phase II clinical status have MWT ≤ 500, log *p* ≤ 5, H-bond donors ≤ 5, and H-bond acceptors ≤ 10 [26]. SwissADME predicts an additional six physicochemical parameters associated with drug-likeness such as lipophilicity [27], size, polarity [28], solubility [29,30], flexibility, and saturation [31,32] as illustrated in Figure 2 and Table 2. The SWISSADME plot of drug-likeness of the isolated compound showed that most of the physicochemical properties of the compound are within the desirable range [33] except for the number of H-bond acceptors, H-bond donors, and polarity of the compound since the PSA value is 201.28 Å^2^ (desirable range (between 20 and 130 Å^2^) [28]. This high polarity may be attributed to the presence of a sugar moiety. The isolated compound showed promising predicted topological aqueous solubility log S (Ali) [29] and Log S (ESOL) [30]. This will facilitate the future formulation of this compound. The predicted pharmacokinetics showed that the compound has low predicted GIT absorption with no permeation to BBB as per the Dania boiled egg model [34] adopted by the SwissADME online server [33]. Furthermore, the compound showed no inhibitor effects on five isoforms of cytochrome P450 (1A2, 2C19, 2C9, 2D6, and 3A4) [35], indicating a low possibility of drug–drug interactions. The predicted skin permeation is calculated by a multiple linear regression model that correlates molecular size and lipophilicity with skin permeability [36]. The more negative the log Kp (with Kp in cm/s), the less skin permeant is the molecule [33,36]. The predicted skin permeation log Kp in cm/s is −9.14 cm/s.

### 2.3. Assessment of Anti-Skin Aging Properties

#### 2.3.1. Determination of Anti-Collagenase and Anti-Elastase Activities

Collagen and elastin are vital structural proteins of the epidermis that sustain the elasticity, capillary tone, and strength of the skin together with hyaluronic acid [39]. During the aging process, oxidative stress, and excessive exposure to UV light lead to the activation of hydrolyzing enzymes such as elastase, collagenase, and hyaluronidase, thereby the skin strength and flexibility are lost, and wrinkles appear [40]. The inhibition of collagenase and elastase activities is one of the effective strategies to protect the skin from skin aging manifestations [41]. As clarified in Figure 3A, the compound showed moderate anti-collagenase property by with IC_50_ value of 419.10 µg/mL, compared to phenanthroline (IC_50_ = 182.80 µg/mL) as a standard.

Regarding the elastase inhibitory effect (Figure 3B), it showed a remarkable effect with an IC_50_ value of 1.06 µg/mL, while the positive control, N-methoxysuccinyl-Ala-Ala-Pro-Val-chloromethyl ketone, exerted an IC_50_ value of 0.63 µg/mL. In accordance with these results, a prior study reported that the methanol leaf extract of mango was a 10-fold potent elastase inhibitor than the standard tocopherol ascribed to the non-competitive inhibition property of mangiferin [42]. Further, previous reports proved the anti-elastase activity of polyphenolic compounds due to the presence of hydrophilic groups such as hydroxyl or carboxyl that could positively influence the competitive inhibition of enzymes [43].

#### 2.3.2. Determination of Anti-Hyaluronidase and Antityrosinase Activities

Hyaluronic acid is the prevalent glycosaminoglycan of the skin that preserves its moisture content [44]. It is enzymatically hydrolyzed by hyaluronidase resulting in a breakdown of the integrity of the skin structure and the disruption of tissue permeability [45]. The suppression of hyaluronidase maintains skin integrity, delays the progression of skin aging, and retains skin hydration [13]. Earlier studies informed that natural hyaluronidase inhibitors could serve well as anti-aging agents for skin health-related products development [46]. As illustrated in Figure 4A, the isolated compound could remarkably attenuate hyaluronidase activity with an IC_50_ value of 1.65 µg/mL, compared to 6-O-palmitoyl L-ascorbic acid as a positive control (2.55 µg/mL). 

Among clinical skin senescence manifestations, skin hyperpigmentation can be accelerated by the tyrosinase enzyme in the presence of reactive oxygen species (ROS) [46]. Tyrosinase, a copper-containing glycoprotein, plays a crucial function in melanin synthesis through hydroxylation of L-tyrosine to 3,4-dihydroxyphenylalanine (DOPA), followed by oxidation of DOPA to DOPAquinone [47]. Therefore, suppression of tyrosinase activity potentially declines skin hyperpigmentation [48]. From this view, the compound potentially inhibited tyrosinase with an IC_50_ value of 0.48 µg/mL, compared to kojic acid as a positive control (0.82 µg/mL) as shown in Figure 4B. A previous study on *Mangifera indica* seed extract showed outstanding anti-tyrosinase and anti-hyaluronidase results correlated with polyphenolic content [47]. On the molecular level, both tyrosine and DOPA contain hydroxyl groups that express as essential proton donors for tyrosinase activation [49]. Likewise, the compound holds four hydroxyl moieties in its skeleton that can behave as competitive inhibitors by reacting as substrates to tyrosinase. 

## 3. Material and Methods 

### 3.1. Plant Material 

The fresh leaves of *Mangifera indica* L. leaves were obtained at the fruiting stage from a private garden in the Abo-Zabal area (N 30°17′43.5336″ E 31°22′28.254), Qualiobya government, Egypt, on 20 July 2021. They were kindly authenticated by Mrs. Treize Labib, the taxonomy specialist at El-Orman Botanical Garden, Giza, Egypt. Voucher specimens with the code PHG-P-MI-362 were placed at the Pharmacognosy Department gallery, Faculty of Pharmacy, Ain Shams University.

### 3.2. Extraction and Chromatographic Isolation

The air-dried leaves of *Mangifera indica* L. (1.56 kg) were percolated in 70% methanol (27 L) for 5 days, then filtered. The filtrate was totally evaporated in vacuo at 47 °C till dryness to produce a dried residue (90.36 g; 5.79 % *w/w*); the yield of extraction was calculated as the following equation: [total weight of dried extract/total weight of fresh plant] × 100. Then, the obtained extract was applied on Diaion HP-20 for fractionation. The gradient elution (water/methanol) was utilized. Five major fractions were obtained: 100% water, 25% methanol, 50% methanol, 75% methanol, and 100% methanol. The 50% methanol-soluble fraction (26.38 g) was exposed to polyamide and eluted initially with 100% water, then gradient eluted with water/methanol (from 100:0 to 0:100, *v/v*) to obtain six subfractions: 100% water, 20%, 40%, 60%, 80% and 100% methanol. The 40% methanol-soluble subfraction (2.36 g) was applied on Sephadex LH-20 using MeOH (isocratic elution). Similar fractions were combined and evaporated to obtain four main subfractions (A1–A4). The subfraction A3 (0.86 g) was applied on preparative thin-layer chromatography (TLC) plates using butanol:acetic acid:water (BAW; 4:1:5) as a mobile phase developer to isolate the compound as a light-yellow amorphous powder.

### 3.3. Nuclear Magnetic Resonance (NMR) Spectrometer

A Bruker Ascend 400/R (Burker Avance III, Fallanden, Switzerland) spectrometer was used at the Center for Drug Discovery, Research, and Development, Faculty of Pharmacy, Ain Shams University.

### 3.4. Mass Spectrometry

A Finnigan LCQ-DECA mass spectrometer (San Jose, CA, USA) connected to a PDA detector was used for mass spectrometric analysis. Samples were dissolved in H_2_O:MeOH as a mixture and injected directly into the HPLC/ESI-MS system. Both negative and positive ESI ionization ion modes were applied under the following conditions: drying and nebulizing gas, N2; capillary temperature, 250 °C; spray voltage, 4.48 kV; capillary voltage, 39.6 V; tube lens voltage, 10.00 V; and full scan mode in mass range *m/z* 100–2000. ESI-MS analysis for the isolated compound was carried out on a Waters Xevo TQD mass spectrometer with UPLC Acquity mode (Milford, CT, USA) at the Center for Drug Discovery, Research and Development, Faculty of Pharmacy, Ain Shams University.

### 3.5. In-Silico Pharmacokinetics Prediction

Lipinski’s rule of five, physicochemical parameters such as lipophilicity, solubility, and pharmacokinetic properties as GIT absorption, distribution, metabolism, and skin permeation of the isolated compound were conducted using the SwissADME online server [36].

### 3.6. Assessment of Anti-Skin Aging Properties 

#### 3.6.1. Determination of Anti-Collagenase Activity

The inhibitory ability of the isolated compound against collagenase activity was assessed using a fluorometric collagenase inhibitor screening kit (BioVision, catalog no. # K833-100) according to the previously described method [50]. The reference control was (1,10)-phenanthroline. The tested compound and reference control were prepared in concentrations of 1, 10, 100, and 1000 ug/mL for the analysis. The tested compound was prepared in a 96-well plate with a flat bottom. First, the collagenase substrate was dissolved in collagenase assay buffer (CAB). The sample for analysis was prepared by mixing the compound with both collagenase and CAB. Preparing inhibitor control samples was performed by mixing the inhibitor ((1,10)-phenanthroline (80 mM)) with the diluted collagenase enzyme and CAB buffer. Enzyme control was prepared by mixing the diluted collagenase with CAB. The CAB buffer was used as the background control. All samples were incubated at room temperature for 15 min. During this time, the reaction mixture was prepared by mixing the collagenase with CAB. In the following step, the reaction mixture was mixed thoroughly with the prepared samples. The fluorescence was measured immediately at 490 nm excitation wavelength and 520 nm emission wavelength using a microplate reader (FilterMax F5, Thermo Fisher). The measurement was conducted in kinetic mode at 37 °C for 60 min. Samples were prepared in duplicate and the collagenase inhibitory effect of the tested compound was calculated applying the following equation: $$\%\;\mathrm{Relative}\;\mathrm{inhibition}=\frac{\mathrm{Enzyme}\;\mathrm{control}-\mathrm{sample}}{\mathrm{Enzyme}\;\mathrm{control}}\times 100$$

#### 3.6.2. Determination of Anti-Elastase Activity

Anti-elastase activity of the compound was investigated using the EnzChek^®^ Elastase Assay Kit (E-12056) according to the previously reported method [51]. The tested compound and reference control were prepared in concentrations of 0.1, 1, 10, and 100 µg/mL for the analysis. The compound was prepared in 96-well plates with clear bottoms for the fluorometric assay. Briefly, elastase enzyme solutions, elastase substrate, and inhibitor control were prepared as recommended. Diluted elastase solution was then added to the wells. Tested compound, inhibitor control, and enzyme control were added to subsequent wells. The samples were mixed on a shaker and then incubated for 5 min at 37 °C. Assay buffer was mixed with the substrate to prepare the fluorometric reaction mix, which was then added and mixed thoroughly with each sample. The fluorescence was measured immediately at 400 nm excitation wavelength and 505 nm emission wavelength using a microplate reader (FilterMax F5, Thermo Fisher, Waltham, MA, USA). The measurement was made in kinetic mode at 37 °C for 30 min protected from light. All tested samples were prepared and assayed in duplicate and the elastase inhibitory effect of the tested compound was calculated applying the following equation: $$\%\mathrm{Relative}\;\mathrm{inhibition}=\frac{\Delta \mathrm{RFU}\left(\mathrm{test}\;\mathrm{inhibitor}\right)}{\Delta \mathrm{RFU}}\times 100$$
where:ΔRFU = a change in relative fluorescence unitsEC = enzyme control

#### 3.6.3. Determination of Anti-Tyrosinase Activity

The inhibitory activities of oils were tested against collagenase using the Abcam^®^ Tyrosinase Inhibitor Screening Colorimetric Assay Kit (catalog # ab204715). The assay was carried out according to the provided manual. The standard tyrosinase inhibitor was kojic acid. The tested compound was prepared in concentrations ranging from 0.1 to 100 µg/mL for the analysis. About 2 μL of provided tyrosinase was mixed with 48 μL of tyrosinase assay buffer. The samples of essential oils or standard (20 μL) were mixed with the enzyme mixture (50 μL) and incubated at 25 °C for 10 min. Then, 30 μL of tyrosinase substrate solution was added to the wells of essential oils and standard. Afterward, the plate was measured spectrophotometrically at a wavelength of 510 nm for 30–60 min at 25 °C.

#### 3.6.4. Determination of Anti-Hyaluronidase Activity

Hyaluronidase activity was evaluated spectrophotometrically by measuring the amount of N-acetylglucosamine formed from sodium hyaluronate [52]. The tested compound and reference control were prepared in concentrations ranging from 0.1 to 100 µg/mL for the analysis. Fifty microliters of bovine hyaluronidase (7900 units mL^−1^, (Sigma, St. Louis, MO, USA) dissolved in 0.1 M acetate buffer (pH = 3.5) was mixed with 100 µL of a designated concentration of the compound dissolved in 5% DMSO and then incubated in a water bath at 37 °C for 20 min. The control group was treated with 100 µL of 5% DMSO instead of the sample. The optical density of the reaction mixture was measured at 585 nm by spectrophotometry.

### 3.7. Statistical Analysis

All experiments were performed in triplicate. The data for the IC_50_ of enzyme inhibitory activity were statistically examined using a one-way analysis of variance (ANOVA). Statistical analyses were performed using GraphPad Prism 5.0.

## 4. Conclusions and Future Perspectives

Plant-derived natural products are in high demand in the global market for the development of new agents for cosmeceutical designs [44]. In this study, a new xanthone compound, 1,3,5,6-tetrahydroxyxanthone-C-4-β-d-glucopyranoside, was isolated from the leaf extract of *M. indica*. Based on enzyme inhibitory assays, this compound could possibly exhibit good anti-collagenase, anti-elastase, anti-hyaluronidase, and anti-tyrosinase properties, thereby conferring a comprehensive attenuating effect against skin aging-related enzymes. Furthermore, an in silico physicochemical parameter and ADME study support its incorporation into topical dosage forms since it displays promising aqueous solubility and reasonable predicted skin penetration. Therefore, this compound could be applied as an auspicious multifunctional bioactive agent for nutraceutical and cosmeceutical formulations. Nevertheless, toxicological and clinical assessments are potentially promising.

## Figures and Tables

**Figure 1 molecules-27-02609-f001:**
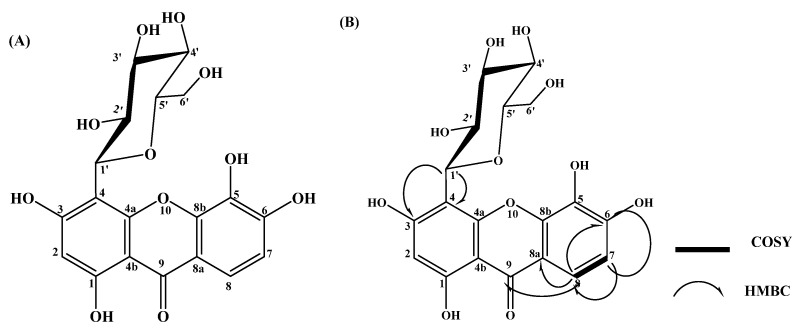
Chemical structure. (**A**) HMBC and ^1^H,^1^H-COSY correlations (**B**) of isolated compound.

**Figure 2 molecules-27-02609-f002:**
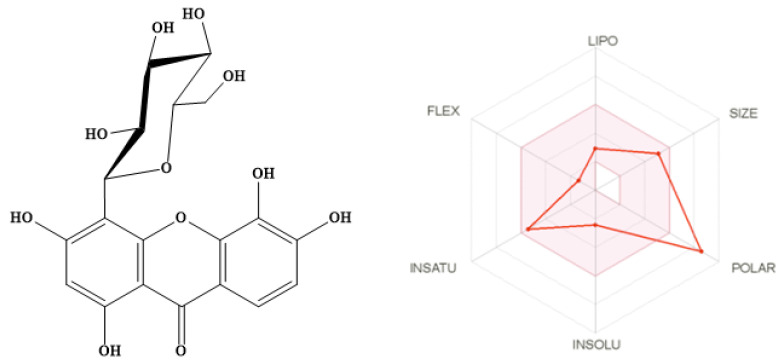
SwissADME plot of drug-likeness of isolated compound. The pink area represents the optimal range for each property (lipophilicity: XLOGP3 between −0.7 and +5.0, size: MW between 150 and 500 g/mol, polarity: TPSA between 20 and 130 Å^2^, solubility: log S not higher than 6, saturation: fraction of carbons in the sp3 hybridization not less than 0.25, and flexibility: no more than 9 rotatable bonds.

**Figure 3 molecules-27-02609-f003:**
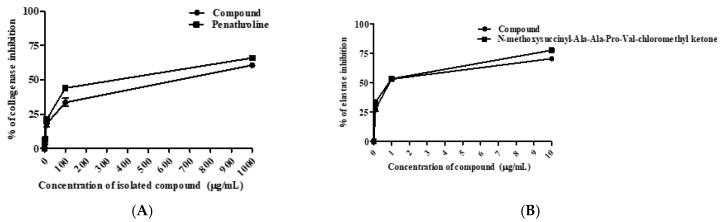
Collagenase (**A**) and elastase (**B**) inhibitory activities of isolated compound. All determinations were carried out in triplicate and the values are expressed as the mean ± SD.

**Figure 4 molecules-27-02609-f004:**
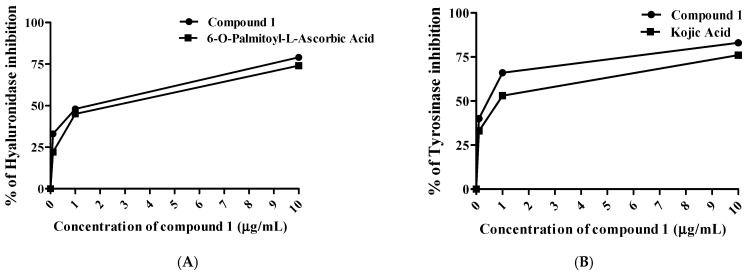
Hyaluronidase (**A**) and tyrosinase (**B**) inhibitory activities of isolated compound. All determinations were carried out in triplicate and the values are expressed as the mean ± SD.

**Table 1 molecules-27-02609-t001:** ^1^H- NMR, APT, and HMBC spectral data of isolated compound (recorded at 400/100 MHz in DMSO-*d_6_*; *δ* in ppm, *J* in Hz).

Position	*δ_H_* (ppm), Multiplicity and *J* (Hz)	*δ_C_* (ppm)	HMBC (H → C)
1	-	1158.95	-
2	5.96, s, 1H	95.35	C-4, 4b, 3
3	-	158.58	-
4	-	104.13	-
4a	-	157.23	-
4b	-	107.45	-
5	-	157.87	-
6	-	161.87	-
7	6.79, d, *J* = 8.24 Hz, 1H	114.80	C6, 8
8	7.57, d, *J* = 8.59 Hz, 1H	131.98	C6, 8a, 9
8a	-	131.22	-
8b	-	158.58	-
9	-	195.14	-
1’	4.60, d, *J =* 7.0, 1H	75.13	C-3, 4
2’	3.21, m, 1H	70.12	-
3’	3.21, m, 1H	78.79	-
4’	3.59, m, 1H	72.34	-
5’	3.21, m, 1H	81.54	-
6’	3.62, dd, 1Ha 3.50, dd,1Hb	60.97	-

**Table 2 molecules-27-02609-t002:** Summary of SwissADME predicted physicochemical descriptors and ADMET parameter of the isolated compound.

Physicochemical Properties			
Molecular weight	422.34 g/mol (≤500) [26]	No. rotatable bonds (not more than 9 rotatable bonds)	2
No. heavy atoms	30	No. H-bond acceptors	11 (H-bond acceptor ≤ 10) [26]
No. arom. heavy atoms	14	No. H-bond donors	8 (H-bond donors ≤ 5) [26]
Saturation: fraction of carbons in the sp3 hybridization	0.32(not less than 0.25) [32]	Topological polar surface area TPSA	201.28 Å^2^ (between 20 and 130 Å^2^) [29]
Lipophilicity: XLOGP3 [30]	−0.37	Solubility	
	(desirable between −0.7 and +5.0)	log S (Ali) [29]	−3.39
		log S (ESOL) [30]	−2.44
Pharmacokinetic properties			
GIT absorption [34]	Low	BBB permeation [34]	No
P-glycoprotein substrate [37]	No	Cytochromes P450 1A2, 2C19, 2C9, 2D6. 3A4 inhibitor [35]	No
Skin permeation (log K_P_) [36]	−9.14 cm/s	Bioavailability score [38]	0.17

## Data Availability

Not applicable.

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
