# Peer review of "A New Xanthone Glycoside from Mangifera indica L.: Physicochemical Properties and In Vitro Anti-Skin Aging Activities"

_molecules, 2022, doi:10.3390/molecules27092609_

Round 1
Reviewer 1 Report
The manuscript “A New Xanthone Glycoside from Mangifera indica L.: Physico-2 chemical Properties and Anti-skin Aging Activities”, shows some interesting results, however, it needs a considerable revision.
Comments:
- Since the authors propose a new "chemical compound", it is necessary to add more analytical evidence, at least the XRD and stereochemistry data.
- Figures 1 and 2, show different stereochemistry!!!
- Are there significant differences in the data presented in figs. 3 and 4?
- What was the negative control?
- In lines 103 and 112, please indicate if the anomeric H, is alpha or beta.
- The references 2,8,10 and 22 are inappropriate self-citations by authors. With all due respect, I suggest to the authors, to eliminate self-citation and leave those that are strictly necessary.
- What is compound TM1 and compound LM1???
- Line 168: what is zhydrolyzing enzymes?
- As suggestion, the term “in vitro” should be added to the title.
- Line 48: In dermo-cosmetics, the term “drugs” is not appropriate.
- Why did you chose compound 1 for the experiments? how many compounds did you isolate?
- Please correct the abbreviation of "grams" according to international standards.
Finally, to continue with the review process, authors should address the above.
Besides addressing the suggestion raised above, with all my respect, I recommend the authors to make more efforts in improving the results and providing more rationale to the discussion.
Reviewer 2 Report
This manuscript describes the physicochemical and anti-skin aging activities of a new compound 1,3,5,6-tetrahydroxyxanthone-C-4-β-D-glucopyranoside. In general, it was well organized and presented promising results. However, there are some aspects that should be improved:
Affiliations and abstract
The affiliations formatation and abstract content in the main document and in the supplementary file were different. Please revise.
In lines 24-25 the phrase should be improved as the formulation and also lipophilicity and solubility are not pharmacokinetic properties. Please revise.
In lines 27-28 again seems to exist some imprecision as the Lipinski rule of five only enables to predict absorption and permeation (and not all the mentioned), maybe you want to refer to the SwissADME server. Please revise.
In addition, in the abstract maybe could be preferably to first describe the ADME results and then the in vitro results following the same order that appears in the manuscript.
1. Introduction
Line 63 ... please delete ..."are that".
Lines 89-90 ... "ADMET" should be only "ADME"
2. Results and discussion
Line 132 - the abbreviation TM-1 was not previously abbreviated!
Line 142- seems to be an error in the abbreviation as was used LM-1 instead TM-1 and also along with different phrases of the manuscript. Please uniformize this abbreviation or use only compound 1.
In table 2 in some cases was inserted the "ideal" values but in order are missing but I think that also could be inserted, in parameters such as, molecular weight, number H-bond acceptors, number H-bond donors, solubility, etc.
In addition, the authors could confirm in the table 2 and in the text if the CYP 2D2 was really this one or it was 2D6? and also for CYP 3A was really this or CYP3A4?
Regarding figure 3 A and B and the corresponding text in lines 165-182 seems to be not consistent. As the authors refer that the collagenase activity was represented in figure 3A but on contrary, in figure 3A it was the elastase activity. in addition, in the figure itself the name of the control compound was cut, and also the legend did not mention that in this figure was also tested this compounds (they are incomplete). Please revise these points.
The number of replicates and how the data was expressed should be indicated in the legends of Figures 3 and 4.
3. Materials and Methods
Regarding the in vitro tests, in all methods should be indicated the concentrations tested of compound 1 and of each control compound.
In addition, all the equations used for the calculation of the results should be indicated properly (with additional explanations of the meaning of each abbreviation).
The references used to cite the in vitro tests (ref.s 53, 54 and 55) are very old techniques in which probably were not used the more recent kits that were applied. Please use more recent ones.
Regarding the statistical analysis, the authors refer that values of p≤0.05 were considered to be statistically significant. However, along with the manuscript and figures and tables, I think that did not observe any indication of p values. There are any statistically significant differences in some test(s)?
4. Conclusion and future prospective should be instead perspectives.
Along the manuscript, there are some other typos and errors that should be corrected and revised.
